# Systematic identification of recognition motifs for the hub protein LC8

Nathan Jespersen[1], Aidan Estelle[1], Nathan Waugh[1], Norman E Davey[2], Cecilia Blikstad[3], York-Christoph Ammon[4], Anna Akhmanova[4], Ylva Ivarsson[3], David A Hendrix[1,5], Elisar Barbar[1]

**Hub proteins participate in cellular regulation by dynamic binding of multiple proteins within interaction networks. The hub protein LC8 reversibly interacts with more than 100 partners through a flexible pocket at its dimer interface. To explore the diversity of the LC8 partner pool, we screened for LC8 binding partners using a proteomic phage display library composed of peptides from the human proteome, which had no bias toward a known LC8 motif. Of the identified hits, we validated binding of 29 peptides using isothermal titration calorimetry. Of the 29 peptides, 19 were entirely novel, and all had the canonical TQT motif anchor. A striking observation is that numerous peptides containing the TQT anchor do not bind LC8, indicating that residues outside of the anchor facilitate LC8 interactions. Using both LC8-binding and nonbinding peptides containing the motif anchor, we developed the "LC8Pred" algorithm that identifies critical residues flanking the anchor and parses random sequences to predict LC8-binding motifs with ~78% accuracy. Our findings significantly expand the scope of the LC8 hub interactome.**

## Introduction

Most proteins interact with few partners, but a class of proteins referred to as hubs interact with a large number of partners in complex protein–protein interaction networks (Jeong et al, 2001, 2000). Hubs can be static or dynamic. Static hubs bind a large number of partners simultaneously at different sites, for example, BRCA2 (Komurov & White, 2007). Dynamic hubs bind multiple partners that compete for the same site (Wu et al, 2009; Patil et al, 2010). Well-known examples of dynamic hubs include calmodulin and 14-3-3 proteins (Aitken, 2006; Uchikoga et al, 2016; Uhart et al, 2016). A more recently discovered member of dynamic hub proteins is the dynein light chain LC8 (Barbar, 2008).

There are more than 280 binary interactions for human LC8 in the Mentha database (Calderone et al, 2013), some of which have been extensively studied, including the dynein intermediate chain (IC) (Makokha et al, 2002; Benison et al, 2006; Nyarko & Barbar, 2011) and the transcription factor ASCIZ (Jurado et al, 2012; Zaytseva et al, 2014; Clark et al, 2018). In addition, expression patterns show that LC8 is highly expressed across a wide variety of cell types (Petryszak et al, 2016) and is broadly distributed within individual cells (Chen et al, 2009; Wang et al, 2016).

LC8 is an 89–amino acid homodimeric protein first identified as a subunit of the dynein motor complex. Colocalization and binding studies with dynein led to a common perception that LC8 functions as a dynein "cargo adaptor" to facilitate transport of dynein cargo (Rodríguez-Crespo et al, 2001; Theerawatanasirikul et al, 2017). However, further studies have shown that LC8 interacts with many proteins not associated with dynein at the same symmetrical grooves in the LC8 dimer interface (Fig 1A). Because of the symmetry of the binding sites of the LC8 dimer, and its association with dimeric proteins, it is now generally accepted that LC8 serves not as a cargo adaptor in the dynein machinery but rather as a dimerization hub in a variety of systems (Barbar, 2008).

LC8 interacts with an 8–amino acid recognition motif within intrinsically disordered regions of its partners. Sequences bound to LC8 form a single β-strand structure integrated into an LC8 antiparallel β-sheet (Clardy et al, 1999; Fig 1). Although there is some variation in the binding motif, it is most frequently anchored by a TQT sequence (Clark et al, 2016). The glutamine in the TQT anchor is typically numbered as position 0 because it is the most highly conserved amino acid (Benison et al, 2007). The flanking threonines are therefore defined as positions −1 and +1. The TQT anchor is highly enriched among known LC8 partners and will be referred to in this article as the "motif anchor" (Fig 1B; Clark et al, 2016).

A dynamic binding interface, determined from nuclear magnetic resonance (NMR) relaxation and hydrogen/deuterium exchange experiments (Fan et al, 2002; Benison et al, 2007; Hall et al, 2008), allows for large sequence variation in LC8 binding partners; however, several steric and enthalpic restrictions are placed on binding sequences. One restriction is inferred from analysis of solvent accessible surface areas of peptides bound to LC8 (Fig 1C;

[1]Department of Biochemistry and Biophysics, Oregon State University, Corvallis, OR, USA   [2]Conway Institute of Biomolecular and Biomedical Sciences, University College Dublin, Ireland   [3]Department of Chemistry - Biomedical Centre, Uppsala University, Uppsala, Sweden   [4]Department of Biology, Utrecht University, Utrecht, The Netherlands   [5]School of Electrical Engineering and Computer Science, Oregon State University, Corvallis, OR, USA

Correspondence: barbare@oregonstate.edu

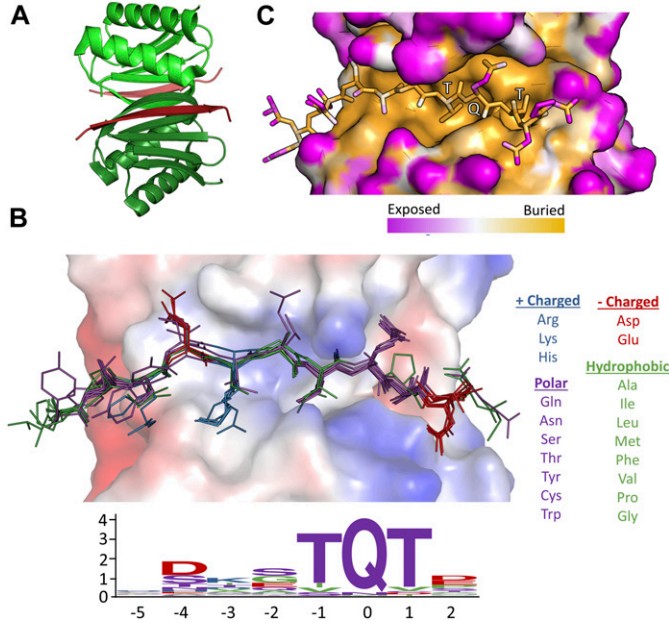

**Figure 1. Motif sequence logo and surface analysis of LC8.**
**(A)** Crystal structure of a representative LC8 dimer (protomers shown in shades of green) bound to a peptide (shades of red). **(B)** Electrostatic charge potential for the LC8 pocket structure using PyMOL's charge-smoothed potential calculator, with positive potentials shown in blue, negative in red, and neutral in white. Peptides from available crystal structures of bound LC8 are shown, and colored based upon amino acid chemical characteristics (right). Amino acid enrichment is shown below each position within the LC8-binding motif, calculated from 79 known binder motifs listed on the LC8 database (http://lc8hub.cgrb.oregonstate.edu). Amino acid letter heights represent relative enrichment of that amino acid. **(C)** Solvent accessible surface area depiction of the same LC8/peptide pair shown in (A). Color scheme was defined at the atomic level using the GetArea program (Fraczkiewicz & Braun, 1998), with magenta representing more solvent exposed and orange regions more buried atoms.

Clark et al, 2016). The side chains of the amino acids at positions –1 and 1 of the peptide (both threonines in Fig 1C) are completely buried, leading to a strong preference for amino acids with branched side chains that are either hydrophobic or, as is the case for threonine, participate in hydrogen bonding. In fact, these two positions are the only side chains that are completely buried (Fig 1C, orange versus pink side chains), suggesting that these residues are under more stringent selective pressures. Interestingly, even though the amino acids on both sides of the anchor are highly variable, their side chains are easily fit within discrete pockets (Fig 1B). In contrast, outside of the 8–amino acid LC8-binding motif, there is higher variability in amino acid sequence and in side chain rotamer conformations (Fig 1B). Analysis of these structures explains the preference for the "TQT" anchor within the LC8 recognition motif but falls short of capturing the spectrum of amino acids that can flank the anchor in potential binding sequences.

In an effort to determine a consensus binding motif, Rapali et al (2011) used phage display and randomized all 8 amino acid positions of the motif except for the conserved glutamine at position 0, and determined VSRGTQTE to be the most thermodynamically favorable binding sequence (Rapali et al, 2011). Although this experiment led to the discovery of multiple LC8 binding partners, the idea of a specific "consensus sequence" belies the dynamics of the

LC8 binding site. In addition, by selecting for the tightest binder, many weaker binders were likely outcompeted and therefore not visible in their study. Our goal in this work is to determine the extent of the variability in LC8 binding sequences flanking the motif anchor.

LC8 motif prediction analyses have increased the number of known binding sequences, and enhanced our understanding of the motif specificity (Erdős et al, 2017; Rapali et al, 2011); however, algorithms generated in these studies were designed for initial screening and are therefore not sufficiently stringent for general use nor made publicly available. Here, we use a combination of proteomic peptide phage display (ProP-PD) technology and position-specific scoring matrices (PSSMs) to determine likely LC8 binding sequences. Interestingly, although our methods were unbiased with respect to the presence of a TQT anchor, sequences experimentally validated to bind LC8 all contained a TQT or variation of the TQT triplet. A database that includes partners identified in this work along with published interactions is now available and contains all 82 validated LC8 interactions. Finally, we used this database to develop an algorithm that incorporates both binding and nonbinding sequences to effectively predict LC8 partners and define rules for LC8 partner recognition that underscore the plasticity of the LC8 binding pocket.

## Results

### LC8 is broadly distributed in cells

To examine the subcellular distribution of LC8, we used HeLa cells stably expressing endogenous levels of a C-terminally tagged LC8-GFP fusion generated by BAC TransgeneOmics (Poser et al, 2008). In interphase cells, LC8-GFP is present throughout the cytoplasm, within the cell nucleus, and enriched in patch-like structures at the cell cortex located in the vicinity of focal adhesions (Fig 2A). In mitotic cells, LC8-GFP is present at the spindle and enriched at the spindle poles and prometaphase kinetochores (Fig 2B). Although the mitotic LC8-GFP localization is consistent with LC8 being part of the cytoplasmic dynein complex (Cianfrocco et al, 2015), the nuclear localization and cortical accumulations in interphase cells are not observed for the other core subunits of cytoplasmic dynein, such as the heavy, intermediate, or light ICs, which were previously tagged with GFP and detected in a similar manner (Splinter et al, 2012).

In agreement with the imaging data, analysis of localization patterns for known LC8 partners performed using the COMPART-MENTS database (Binder et al, 2014) show that they can be found in multiple cellular compartments, such as the cytoplasm, nucleus, and vesicles (Fig 2C). Surprisingly, LC8 partners can even be found in extracellular space. Localization within the various categories of subcellular structures supports the conclusion that LC8 is broadly distributed throughout the cell irrespectively of dynein and concentrates at certain subcellular sites where LC8 partners are enriched.

### ProP-PD selections identify 16 new LC8 interactions

For broad mapping of the LC8 interaction network, we used a ProP-PD assay, in which a library is created that encodes sequences for

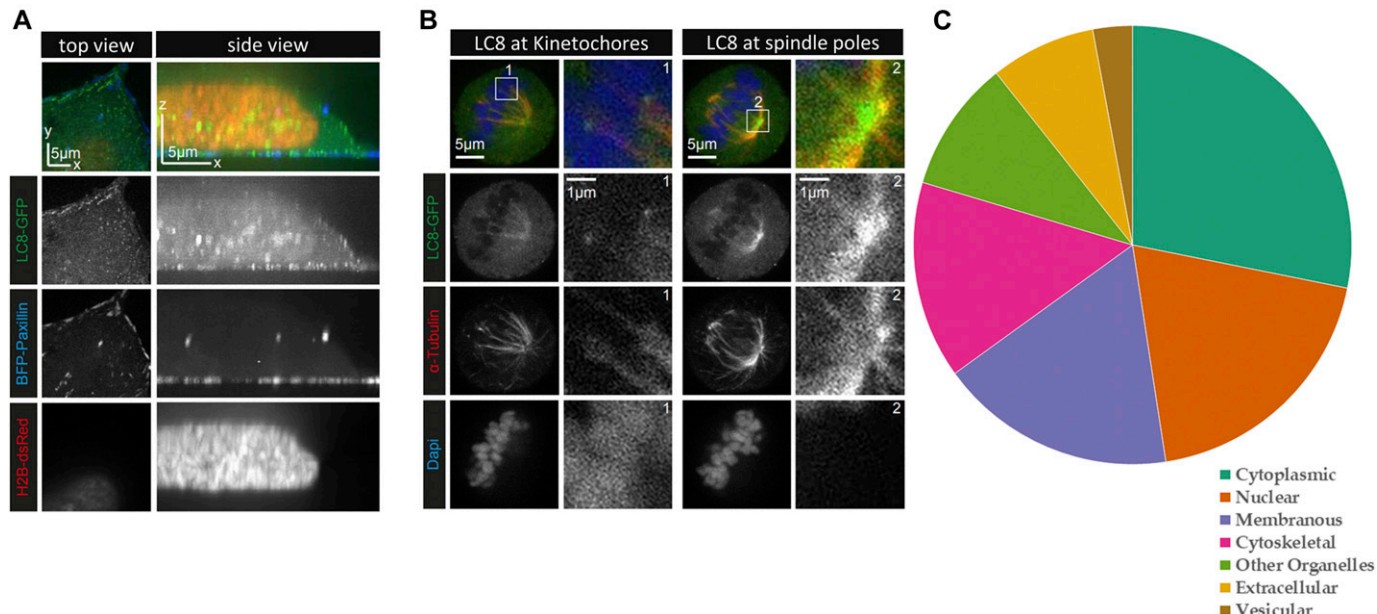

**Figure 2. LC8 and its binding partners display broad cellular localization.**
**(A)** Live HeLa cells stably expressing LC8-GFP (green) were transiently transfected with the focal adhesion marker TagBFP-paxillin (blue) and the nucleus marker dsRed-tagged histone H2B (red). The top view images shown on the left represent an optical section located next to the coverslip. LC8-GFP is present throughout the cell but forms puncta at the cell cortex. **(B)** HeLa cells stably expressing LC8-GFP were fixed with PFA and stained for the endogenous α-tubulin (red) and with DAPI (blue) to visualize the DNA. LC8 accumulates at the kinetochores (box 1) and at the spindle poles (box 2). In both (A) and (B), images were acquired from a single cell each, using confocal spinning disc microscopy. **(C)** Localization information derived from the COMPARTMENTS program demonstrates that LC8 binding partners are localized to all cellular compartments. High confidence localization data were available for 59 of the 73 eukaryotic proteins listed on the LC8Hub database.

peptides drawn from a genome of interest. These peptides are then synthesized by bacteriophages and displayed externally on bacteriophage coats. Our ProP-PD library is composed of 479,846, 16–amino acid–long peptides with overlapping segments designed from the disordered regions of 18,692 human proteins (Davey et al, 2017). Through phage display selections using immobilized LC8 as bait protein, we pulled down and sequenced 53 potential binding partners with highly variable sequences, including five previously identified partners (Table S1).

To validate the binding of these partners, we synthesized 14–amino-acid–long peptides containing the motif anchor at positions 10–12 and tested their binding by isothermal titration calorimetry (ITC). Of the 53 synthesized peptides, 16 interacted with LC8 to varying degrees, whereas 31 peptides showed no binding under our experimental conditions (Table 1). Binding of the remaining peptides was not tested because of their poor solubility.

Binding dissociation constants ranged from 0.16 $\mu$M for BCL2L11, to affinities too weak to be reliably determined by ITC ($K_d$ greater than 25 $\mu$M cutoff). Representative thermograms for strong binders (data fit with high accuracy), weak binders (not fit), and nonbinders are shown in Fig 3A. Under our experimental conditions, LC8 is a dimer (Barbar et al, 2001); therefore, the 1:1 binding ratio in all ITC thermograms corresponds to two peptides binding to an LC8 dimer, as expected. Although the ProP-PD method has no innate bias toward anchor-containing sequences, most of the LC8-binding peptides contain the TQT sequence (Table 1). However, many anchor containing peptides failed to bind in our ITC assays, suggesting that the anchor motif is strongly predictive of high-affinity binding but insufficient to guarantee interactions with LC8.

There are several plausible explanations for the lack of binding of ProP-PD–derived peptides to LC8. In the hybrid M13 system used here, each phage particle displays hundreds of peptide copies on its surface. This avidity allows the capture of low-/medium-affinity interactions, in the range of 40–150 $\mu$M (Davey et al, 2017; Wu et al, 2017). In addition, because LC8 binds much more tightly to dimeric partners than to monomeric partners because of the bivalency effect (Hall et al, 2008), the presence of multiple peptides in close proximity on the phage surface could facilitate binding to otherwise weak motifs. Thus, it is possible that the anchor-containing peptides that failed to bind in our ITC experiments, in fact do bind LC8, but with affinities weaker than 40 $\mu$M.

Available crystal structures of LC8 bound to partner peptides indicate that partners require amino acids on the N-terminal of the TQT anchor motif for the necessary backbone H-bonds to form a $\beta$-strand. Therefore, for the phage display identified peptides that had the TQT anchor at the N-terminus, the corresponding synthesized peptide sequence was shifted to have the TQT motif at the 10–12 position, and the flanking regions were replaced with the actual protein sequence at these positions to allow for at least 5 amino acids on the N-terminal side of the TQT sequence.

This design may partially explain why some peptides that were pulled down in the ProP-PD experiments did not bind via ITC, as the sequences used in these experiments were not identical. Many of the nonbinding ProP-PD hits have their TQT anchor located near the N-terminus of the peptide. As each ProP-PD peptide is N-terminally flanked by an SSSG linker, their binding behavior might be expected to differ from native sequences, wherein flanking regions are different

**Table 1. Peptides synthesized based on their binding in phage display experiments.**

| UniProt | Gene | Sequence | Start | End | Binder? | Citation |
|---|---|---|---|---|---|---|
| O43521 | BCL2L11 | APMSCDKSTQTPSP | 108 | 117 | Y | Puthalakath et al (1999) |
| Q9UPA5 | BSN[II] | PRATAEFSTQTPSP | 1,498 | 1,511 | Y | Fejtova et al (2009) |
| Q86VQ1 | GLCCI1 | SSSTRSIDTQTPSV | 340 | 353 | Y | Hutchins et al (2010) |
| Q96R06 | SPAG5 | HPETQDSSTQTDTS | 463 | 476 | Y | Schmidt et al (2010) |
| Q6IMN6 | CAPRIN2 | NQSFTTASTQTPPQ | 791 | 804 | Y | |
| O75665 | OFD1 | AKESCNMETQTSST | 153 | 166 | Y | Boldt et al (2016) |
| Q02505 | MUC3A | PVLTSATGTQTSPA | 1,800 | 1,813 | Y | |
| Q9UBY0 | SLC9A2 | *DD*HSREKGTQTSG*D* | 749 | 804 | Y | |
| Q9Y2F5 | ICE1 | EKELRHIGTQISSD | 181 | 194 | Y | |
| Q9ULV3 | CIZ1 | ARAGRSVSTQTGSM | 13 | 26 | Y | den Hollander & Kumar (2006) |
| Q99102 | MUC4 | SQNHWTRSTQTTRE | 200 | 213 | Y | |
| Q99102 | MUC4 | *DD*NHWTRSTQTTRE | 200 | 213 | Y | |
| P07359 | GP1BA | GQGAALTTATQTTHLE | 566 | 581 | Weak[a] | |
| Q9Y4F4 | TOGARAM1 | SKTQQTFGSQTECT | 788 | 801 | Weak[a] | |
| Q8WWN8 | ARAP3 | SPSPTGLPTQTPGF | 1,514 | 1,527 | Weak[a] | |
| Q01973 | ROR1 | *DD*SGGNATTQTTS*D* | 760 | 774 | Weak[a] | |
| Q8NEZ4 | KMT2C | IVSCVSVSTQTASD | 205 | 218 | N | |
| Q9UPA5 | BSN-shift[b] | STQTPSPAPASDMP | 1,505 | 1,518 | N | |
| Q7Z2Z2 | EFL1 | *D*ERLMCTGSQTFD*D* | 375 | 386 | N | |
| Q02817 | MUC2 | TPTPTPTGTQTPTT | 2,000 | 2,013 | N | |
| Q9HC84 | MUC5B-3 | SMATPSSSTQTSGT | 2,673 | 2,686 | N | |
| Q8TEC5 | SH3RF2 | TLVSTASGTQTVFP | 714 | 727 | N | |
| Q9P2G1 | ANKIB1 | RGDGSDVSSQTPQT | 1,065 | 1,078 | N | |
| O43526 | KCNQ2 | *DD*PMYSSQTQTYG*D* | 370 | 380 | N | |
| P14859 | POU2F1 | ESGDGNTGTQTNGL | 13 | 26 | N | |
| P35568 | IRS1 | LPRKVDTAAQTNSR | 841 | 854 | N | |
| Q2KHR3 | QSER1 | KTLTFSGSSQTVTP | 374 | 387 | N | |
| Q99814 | EPAS1 | TEAKDQCSTQTDFN | 509 | 522 | N | |
| Q9Y4K1 | CRYBG1 | RSFVLPVESTQDVSSQ | 550 | 565 | N | |
| P49862 | KLK7 | SFRHPGYSTQTHVN | 98 | 111 | N | |
| Q92904 | DAZL | TQDDYFKDKRVHHFRRS | 272 | 288 | N | |
| Q96FV2 | SCRN2 | VRTLPRFQTQVDRR | 342 | 355 | N | |
| Q96FV2 | SCRN2 | *DD*TLPRFQTQVDRR | 344 | 355 | N | |
| Q7Z589 | EMSY | KITFTKPSTQTTNT | 261 | 274 | N | |
| Q13952 | NFYC | CLKETLQITQTEVQ | 289 | 302 | N | |
| Q9HC84 | MUC5B-1 | TTLPVLTSTATKST | 3,049 | 3,062 | N | |
| P53350 | PLK1 | AASLIQKMLQTDPTAR | 278 | 293 | N | |
| Q92499 | DDX1 | *DD*HSGNAQVTQTKF*D* | 271 | 282 | N | |
| Q8NBH2 | KY | ITSYNSQGTQLTVE | 81 | 94 | N | |
| Q06190 | PPP2R3A | LQETLTTSSQANLS | 625 | 638 | N | |
| Q13618 | CUL3 | KHSGRQLTLQHHMG | 542 | 555 | N | |
| Q9H4B6 | SAV1 | NQSFLRTPIQRTPH | 70 | 83 | N | |
| Q2TV78 | MSTL1 | EGYRGTANTTTAAYLA | 259 | 274 | N | |

**Table 1. Continued**

| UniProt | Gene | Sequence | Start | End | Binder? | Citation |
|---------|------|----------|-------|-----|---------|----------|
| Q6ZU65 | UBN2 | PLQATISKSQTNPV | 942 | 955 | N | |
| Q96SC8 | DMRTA2 | SSRSAFSPLQPNAS | 433 | 446 | N | |
| Q6ZRI0 | OTOG | TLQQPLELTASQLPAG | 1,541 | 1,556 | N | |
| Q96JG9 | ZNF469 | RAAALPEETRSSRR | 1,014 | 1,027 | N | |

Anchor motifs are underlined.
Aspartates shown in italics were added to increase solubility.
[a]Peptides that displayed an interaction with LC8 via ITC, but the data were not of sufficient quality to obtain reliable $K_d$ measurements.
[b]BSN-ProP-PD is the Bassoon sequence pulled down by phage display, without shifting the TQT sequence into the correct position.

and might contain amino acids that impair LC8 binding. This effect is most clearly demonstrated by our results for two partially complementary peptides derived from the protein bassoon (BSN). Two versions of this peptide were synthesized for this experiment: one version exactly matching the ProP-PD hit with the TQT motif at the N-terminus (minus the SSSG tag, STQTPSPAPASDMP) and a modified version with the TQT motif near the C-terminus (PRATAEFSTQTPSP). The modified BSN peptide with a C-terminal TQT demonstrated strong binding to LC8, whereas the original phage-determined peptide with N-terminal TQT failed to demonstrate even weak binding to LC8. This example provides strong evidence that LC8–partner binding interactions are sensitive to the positioning of the TQT anchor within the full binding motif and validate our concern that the SSSG linker erroneously facilitated binding for some sequences in the ProP-PD experiment.

### Position-specific scoring and disorder prediction identifies seven new LC8 interactions

We scanned *Homo sapiens* and viral proteomes for potential LC8-binding sequences based upon: (1) their propensity for disorder, (2) their sequence conservation across related species, and (3) the sequence's similarity to known motifs. This final parameter used a PSSM based on the relative enrichment or depletion of each residue in a potential sequence, weighted by amino acid frequency. Enrichment and depletion PSSMs were populated using sequences that interact with LC8, verified either by mutagenesis or in vitro assays. ITC experiments were performed on 19 synthetic peptides with high-scoring sequences, but only seven bound (Table 2). These include the human papillomavirus E4 protein, the rotaviral VP4, and human CCL2, CCL7, SON, MAST1, and ZFPM1 proteins. Notably, a previous study also predicted a binding site within the rotavirus A VP4 protein (Martínez-Moreno et al, 2003) at residue position 644–651 (IDMSTQIG); however, the synthesized peptide sequence tested in that study did not bind LC8. Here, we predict an alternative site at positions 605–612 (NDISTQTS) based on disorder propensity and motif similarity. ITC experiments confirmed that the site predicted here binds with a $K_d$ of 4.2 $\mu$M (Fig 3B).

### Common motif features that promote LC8 binding

Because only 7 of the 19 PSSM-predicted binding sequences actually interact with LC8, it is clear that an additional filtering method should be introduced to minimize false positives. To assess common features for binding from this growing dataset of

interactions, we overlaid all known tight binding partners (50 sequences with $K_d$s <10 $\mu$M, Fig 4B) and all nonbinding sequences (determined here, Fig 4C). This comparison revealed some conspicuous differences between binders and nonbinders, allowing for the determination of the position-based rules that follow (Fig 4A and D).

The anchor is extremely well conserved in both amino acid type and volume. There is a strong preference for a mid-sized H-bonding/hydrophobic residue at positions −1 and +1 and a clear preference for a glutamine at position 0. Any deviation from this anchor, such as the RQT seen in EIF4G3, leads to a nonbinding sequence. Both threonines are completely buried in crystal structures (Fig 1C), and therefore, deviations to a charged group are highly unfavorable (Fig 4D, Poor Anchor).

Position +2, which has no $\beta$-strand backbone interactions in any crystal structures, shows a large preference for proline, aspartate, and glutamate residues. Interestingly, these three residues are classically depleted in $\beta$-strands (Chou and Fasman, 1974; Kumar, 2013), providing a potential explanation for these residues acting as "strand-breaking" amino acids at the periphery of the LC8 binding pocket. An alternative explanation for their enrichment is that the negative charge for E and D can interact with the positive electrostatic charge on LC8 (Fig 1B). Proline, however, might energetically assist in binding by reducing the change in entropy, as both proline and pre-proline residues are conformationally restricted (Ho & Brasseur, 2005). Hydrophobic amino acids are not well accommodated at this position (Fig 4D, Hydrophobic +2).

Position −2 shows little charge preference and allows positive, negative, polar, and hydrophobic residues; however, there are no examples of bulky aromatic side chains at this position among the tightly binding peptides, indicating that there are some steric constraints (Fig 4D, Bulky Hydrophobic −2).

Position −3 favors large side chains as nearly all tight binders contain an amino acid at least as large as valine at this position, with only two occurrences of an alanine. Fig 1B reveals a binding pocket where large side chains can fit, which is often occupied by lysines or arginines. A small side chain at the −3 position does not immediately exclude a sequence from binding, as in CAPRIN2 (Table 1), but seems to be less favorable based on the depletion of these residues (Fig 4D, Small −3).

Position −4 favors amino acids capable of making a polar contact, such as aspartate, and no sequences identified to date have hydrophobic residues larger than alanine at this position (Fig 4D, Hydrophobic −4). Finally, the −5 position shows a slight bias toward positively charged residues (Fig 4B and C), but it is unclear whether this effect is significant.

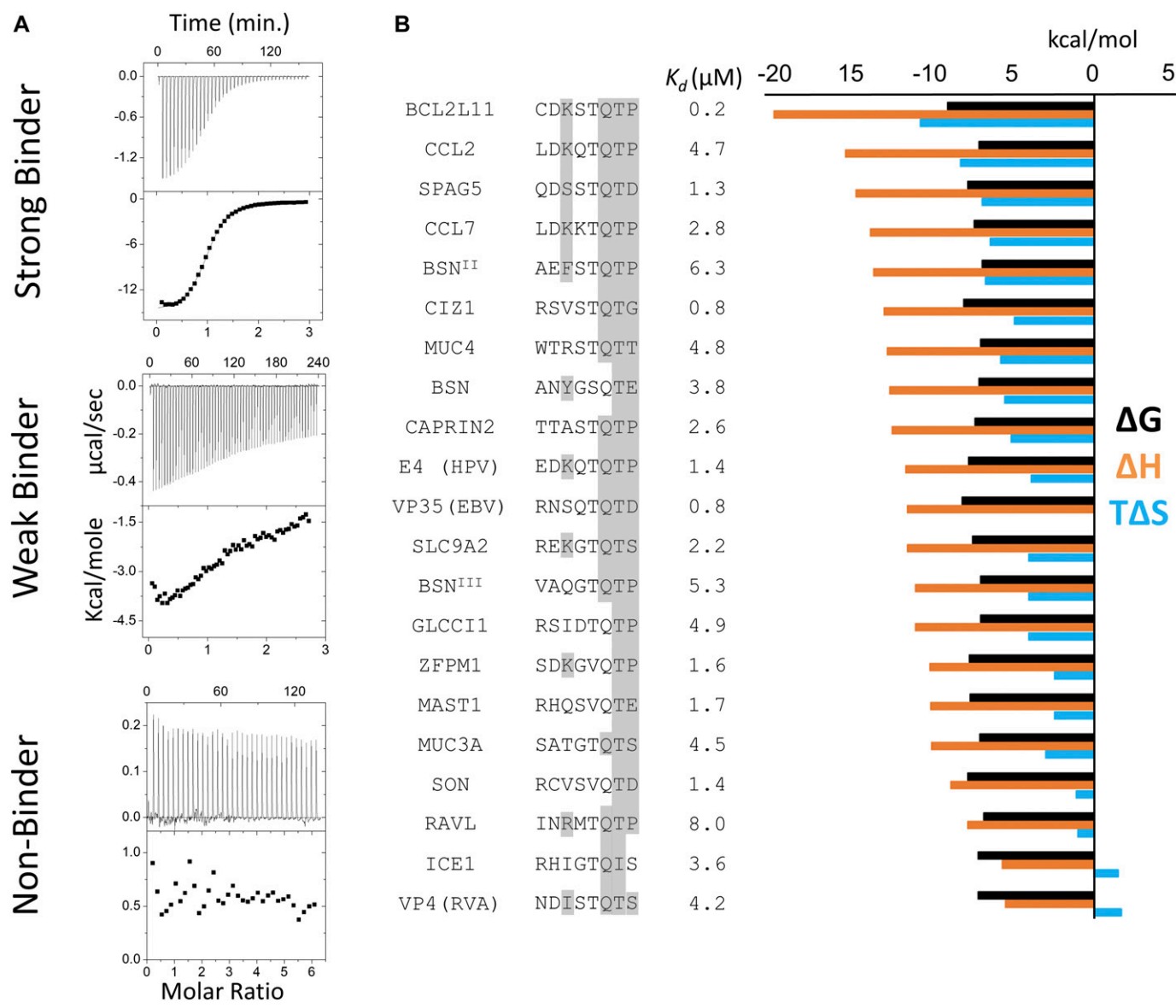

**Figure 3. Thermodynamic analysis of binding peptides.**
**(A)** Representative ITC thermograms of LC8 bound to 14–amino acid–long synthetic peptides. The data were collected at 25°C in 50 mM NaCl, 50 mM NaPO4, and 1 mM NaN$_3$, pH 7.5. Experiments were performed in triplicate. We categorized peptides as strong binders ($K_d$ reliably determined; SPAG5), weak binders (heat generated, but unable to fit the data; TOGARAM1), and nonbinders (QSER1). Weak binders are those with affinities >25 μM. **(B)** Binding affinities and thermodynamic parameters for strong LC8 binders identified in this study. Thermodynamic parameters for all binding peptides at 25°C are shown. ΔG (black), ΔH (orange), and TΔS (blue) kcal/mol values are the average of two to three independent ITC experiments. $K_d$s are shown in μM. 8 amino acid motifs are shown, with residues capable of making conserved hydrogen bonding interactions highlighted in grey. Sequences are ordered by descending ΔH values.

In general, partners must bind within a deep hydrophobic pocket and form a β-strand structure; therefore, multiple similar charges within a peptide, or sterically challenging prolines at any internal position, makes binding unfavourable. Even with this systematic comparison, a number of the nonbinding sequences could not be categorized (Fig 4D).

## The partner-binding pocket is conserved in LC8 sequences but is structurally variable

A comparison of LC8 amino acid sequences from 58 different eukaryotic species using the ConSurf program (Ashkenazy et al, 2016) reveals that the partner binding site is strictly conserved across these diverse organisms (Fig 5A). Interestingly, the conservation of residues creates a noticeable gradient pattern that radiates out from the dimeric interface/partner binding site, with the most conserved residues near the core (maroon), and the least conserved residues at the peripheries (blue).

We used the Ensemblator program (Brereton & Karplus, 2018), which aligns independently determined 3D structures and identifies regions of structural conservation or plasticity, to visualize how a sequence that is strictly conserved is capable of binding such a wide variety of sequences. By overlaying the protomers from five

**Table 2. Peptides synthesized based on favorable sequence comparisons.**

| UniProt | Gene | Sequence | Start | End | Binder? | Citation |
|---|---|---|---|---|---|---|
| Q6LCS3 | E4 (HPV) | YLQGRQEDKQTQTPPP | 16 | 30 | Y | |
| Q8IX07 | ZFPM1 | PAPPSYSDKGVQTPSK | 947 | 962 | Y | |
| P11193 | VP4 (Rotavirus A) | YVTNSLNDISTQTSTI | 600 | 614 | Y | |
| P13500 | CCL2 | YDSMDHLDKQTQTPKT | 85 | 99 | Y | |
| P18583 | SON | YSRKSRCVSVQTDPT | 87 | 100 | Y | |
| Q9Y2H9 | MAST1 | YGCTRHQSVQTEDG | 1,387 | 1,399 | Y | Navarro-Lérida et al (2004) |
| P80098 | CCL7 | QDFMKHLDKKTQTPKL | 84 | 99 | Y | |
| Q5K4E3 | PRSS36 | YGPDGEETETQTCPP | 468 | 581 | N | |
| P20702 | ITGAX | YGQIAPENGTQTPSP | 1,146 | 1,159 | N | |
| P03586 | MT/HEL (TMV) | AQPKQKLDTSIQTEYP | 1,305 | 1,320 | N | |
| Q8IYH5 | ZZZ3 | KSVAENGDTDTQTSMF | 237 | 252 | N | |
| Q5DMI6 | DNLJ2 (phage T5) | YKIEIPTQCPSCGSK | 2 | 15 | N | |
| Q92904 | DAZL | YPQKKSVDRSIQTVVS | 243 | 257 | N | |
| Q9NZ56 | FMN2 | YHHRILEAKSIQTSPT | 735 | 749 | N | |
| Q13418 | ILK | MDDIFTQCREGN | 1 | 12 | N | |
| O43432 | EIF4G3 | DFTPAFADFGRQTPGG | 676 | 691 | N | |
| Q99613 | EIF3C | YELMASLDQPTQTVVM | 830 | 844 | N | |
| O15444 | CCL25 | NKVFAKLHHNTQTFQA | 94 | 109 | N | |
| P20042 | EIF2S2 | KPFMLDEEGDTQTEET | 21 | 36 | N | |

Anchor motifs are underlined.
Tyrosines shown in italics were added for accurate concentration determination.

published crystal and NMR structures of free LC8, we observed that the β-strand that directly binds to partners is highly variable (Fig 5B; Fan et al, 2002; Benison et al, 2007; Hall et al, 2008) and has the highest root mean squared deviation (RMSD) values between structures. It is of note that the most sequence-conserved region is also the most structurally variable part of the protein. This structural plasticity allows accommodation of a diverse set of partners with a wide range of properties and sheds light on why definitive identification of LC8-binding motifs is such a difficult task.

### Enthalpic and entropic modes of binding

Crystal structures of LC8 bound to different peptides reveal surprisingly few conserved backbone and side chain H-bonds. Backbone H-bonding between the antiparallel β-strands occurs only for residues between positions −5 and −1 (Fig 5C and D). Tellingly, there are only five frequently observed side chain H-bonds, and four of these occur within the anchor (Fig 5E). The remaining interaction occurs at position −4, which is often an aspartate residue. The conspicuous lack of conserved side chain polar contacts, along with LC8's dynamic binding interface, suggests that most disordered (or extended) anchor-containing sequences should be capable of binding LC8.

Thermodynamic data obtained by ITC demonstrate that all of our peptides bind to LC8 in an enthalpically driven reaction (Fig 3B). Analysis of all peptides with strong anchors, high binding affinities, and similar $K_d$s shows large differences in their ΔH and TΔS values,

spanning 15 kcal/mol. In general, peptide sequences that contain TQT anchor sequence and the capacity for a polar contact at the −4 position have the largest ΔH (e.g., BCL2L11 and CCL2), whereas sequences like MAST1 or ICE1 that lack one or more of these interactions have a lower ΔH (Fig 3B). Some outliers (such as VP4, BSN, and RavL) have lower ΔH values. One possible explanation is that these peptides have predicted helical structures, which require additional energy to unfold before binding and thus result in smaller overall ΔH values. Indeed, the LC8 recognition motif within VP4 has the smallest ΔH (−5.6) and is predicted by IUpred to be ordered (average disorder propensity of 0.37).

### Incorporation of physicochemical features and nonbinder data improves binding predictions

Based on position preferences described above, we developed an LC8Pred algorithm that captures common features observed in binding peptides, including size and charge preferences, and features present in the 32 anchor-containing nonbinding peptide sequences (Fig 6A). For each matrix, positive values within the matrix indicate that the given amino acid is enriched in binding sequences and depleted in nonbinding sequences, whereas high negative values signify depletion of that amino acid in binding sequences and enrichment in nonbinding sequences. The addition of nonbinding sequence information significantly improved the algorithm's capacity to differentiate between binding and nonbinding sequences; however, with only 32 nonbinding sequences,

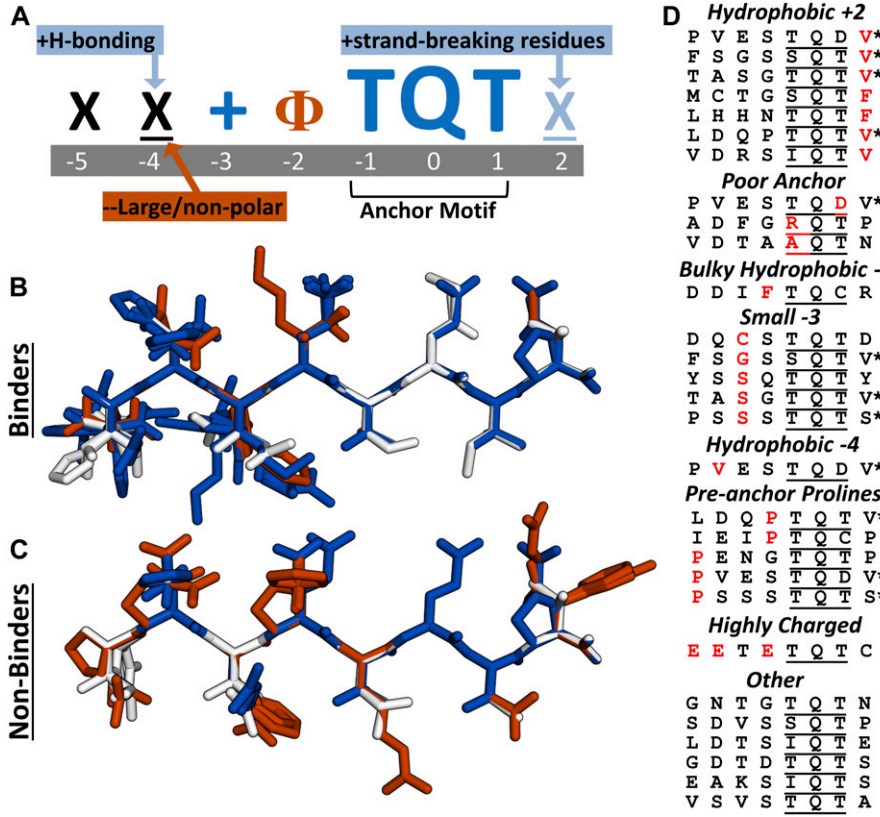

**D**

**Hydrophobic +2**

| | | | | | | | |
|---|---|---|---|---|---|---|---|
| P | V | E | S | T | Q | D | V* |
| F | S | G | S | S | Q | T | V* |
| T | A | S | G | T | Q | T | V* |
| M | C | T | G | S | Q | T | F |
| L | H | H | N | T | Q | T | F |
| L | D | Q | P | T | Q | T | V* |
| V | D | R | S | I | Q | T | V |

**Poor Anchor**

| | | | | | | | |
|---|---|---|---|---|---|---|---|
| P | V | E | S | T | Q | D | V* |
| A | D | F | G | R | Q | T | P |
| V | D | T | A | A | Q | T | N |

**Bulky Hydrophobic -2**

| | | | | | | | |
|---|---|---|---|---|---|---|---|
| D | D | I | F | T | Q | C | R |

**Small -3**

| | | | | | | | |
|---|---|---|---|---|---|---|---|
| D | Q | C | S | T | Q | T | D |
| F | S | G | S | S | Q | T | V* |
| Y | S | S | Q | T | Q | T | Y |
| T | A | S | G | T | Q | T | V* |
| P | S | S | S | T | Q | T | S* |

**Hydrophobic -4**

| | | | | | | | |
|---|---|---|---|---|---|---|---|
| P | V | E | S | T | Q | D | V* |

**Pre-anchor Prolines**

| | | | | | | | |
|---|---|---|---|---|---|---|---|
| L | D | Q | P | T | Q | T | V* |
| I | E | I | P | T | Q | C | P |
| P | E | N | G | T | Q | T | P |
| P | V | E | S | T | Q | D | V* |
| P | S | S | S | T | Q | T | S* |

**Highly Charged**

| | | | | | | | |
|---|---|---|---|---|---|---|---|
| E | E | T | E | T | Q | T | C |

**Other**

| | | | | | | | |
|---|---|---|---|---|---|---|---|
| G | N | T | G | T | Q | T | N |
| S | D | V | S | S | Q | T | P |
| L | D | T | S | I | Q | T | E |
| G | D | T | D | T | Q | T | S |
| E | A | K | S | I | Q | T | S |
| V | S | V | S | T | Q | T | A |

**Figure 4. Analysis of LC8-binding and nonbinding motifs reveals distinct positional preferences. (A)** Motif preferences for LC8 binding partners. "Φ" denotes hydrophobic residues; "X" signifies any residue (unless certain residues are disfavored); underlined "X" signifies any residue but with strong preferences for particular residues; "+" denotes positively charged amino acids. Physiochemical properties beneficial for binding are colored dark blue or light blue, based on magnitude, and deleterious properties are colored in red. **(B)** All known tightly binding sequences ($K_d <$ 10 $\mu$M) are cropped to 8 amino acid motifs and built using the Chimera molecular modeling software. This includes LC8 sequences found on the LC8Hub database, and those determined in this article. **(C)** Overlay of all nonbinding peptides used in this study. Residues are colored based upon whether they are beneficial (blue), deleterious (red), or neutral (white) for binding, using the amino acid enrichment and depletion in known motifs (Fig 6A). **(D)** Categories of nonbinding sequences. Residues highlighted in red depict the reason the sequence is placed within a given category. *Denotes sequences placed in multiple categories.

our data were notably sparse, and separation between the two groups was incomplete. To improve our differentiation capacity, we binned the 20 amino acids into four categories and developed additional PSSMs using these bins, thereby reducing the overall number of matrix terms. The first PSSM separated amino acids based on polarity, whereas the second PSSM separated according to volume.

These matrices largely confirm groupings as described in the "common feature" section above, but with some exceptions. Notably, although there is a preference for large amino acids at the –3 position, the polarity matrix also shows an enrichment in positively charged residues. In addition, although the –5 position is the most varied in the matrix, it has a high score for positively charges residues (Fig 6A). This discrepancy is because of the lack of positively charged residues at –5 in the nonbinding sequences rather than from any strong enrichment of positive charge in the binding sequences. The –5 position also shows a slight enrichment for very large amino acids and is the only position to do so. Crystal structures show that the –5 position is not buried within LC8's binding groove and therefore experiences much less steric restriction (Fig 1B).

Using the described matrices, we scored all known binders and nonbinders to determine the discriminatory capabilities of the PSSMs (Fig 6B). Although the amino acid, volume, and polarity matrices were each moderately successful at separating binding from nonbinding sequences in isolation, the best separation was achieved when every matrix was combined. We combined the

volume and polarity matrices to determine a volume and polarity score, and the amino acid matrix was used to determine an amino acid score (Fig 6B).

Because our goal is to predict partners with high reliability, strict thresholds were used to determine what constitutes a binder and a nonbinder. A minimal score of 12.9 on the amino acid matrix, and 0.1 on the volume and polarity matrix, is used to determine whether a sequence is likely to be considered a binder. These thresholds result in only four false positives and 20 false negatives with our available data set, corresponding to a 75% true-positive rate and an 88% true negative rate (Fig 6B). Interesting, although the volume and polarity matrices only provide a small increase in accuracy overall at these thresholds, they are extremely proficient at separating binders from nonbinders when applied stringently. A threshold of 2.7 on the volume and polarity matrix alone results in a 0% false-positive rate, while retaining 57% of the true positives (Fig 6B).

Although we achieve an accuracy of 78%, there are a number of outliers: both high-scoring nonbinders and low-scoring binders. Within the binders, the first sequence, DDKNTMTD, is from Myosin Va (Fig 6B). It is unsurprising that this sequence scores poorly, as it is the only "TMT" anchor with verified binding data, and therefore has a low score because of the M instead of Q. However, binding is likely salvaged by the presence of the highly favourable amino acids at the other positions and by the presence of adjacent coiled-coil domains in the full-length protein. The remaining three lowest scores belong to proteins with multiple LC8-binding sequences

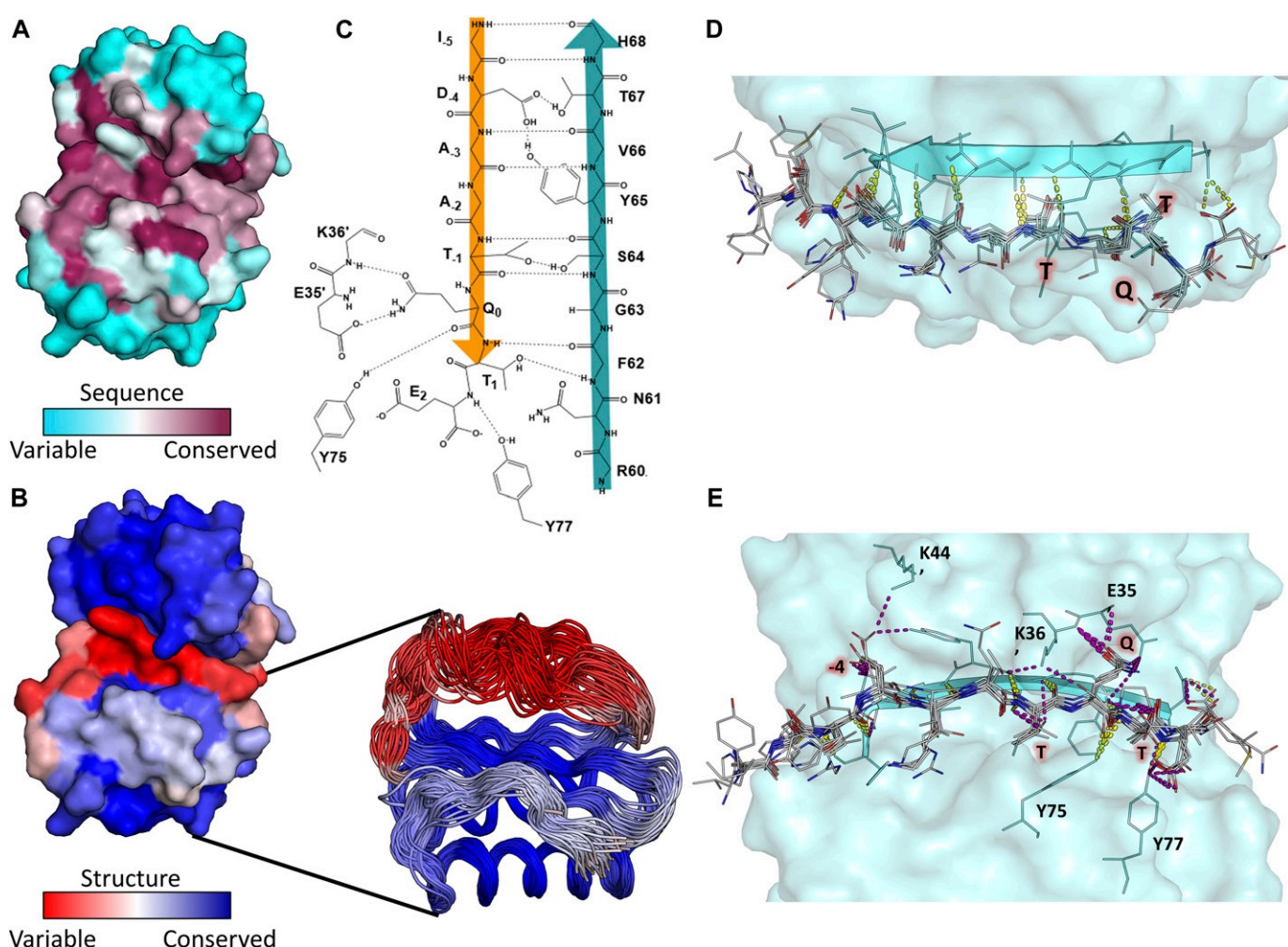

**Figure 5. LC8 is structurally variable but conserved in sequence.**
**(A)** Surface representation of LC8 colored by sequence conservation using ConSurf. More sequence-conserved regions are shown in magenta, less sequence-conserved regions are shown in cyan. Highly conserved residues map to those within the LC8 binding site. **(B)** Surface representation of LC8 colored by *structural* conservation in the free protein using the Ensemblator. Regions that are more structurally variable are shown in red, whereas more structurally conserved regions are shown in blue. An overlay of NMR and crystal structure protomers used for the structural analysis is shown as a cut-out in (B). **(C)** 2D depiction of the binding interface between an example peptide (orange) and the binding β-strand within LC8 (Teal). **(D, E)** Polar bonds between LC8 and peptides from crystal structures are shown in (D) (top down view, only backbone interactions) and (E) (pocket view). Colors of polar contacts are based on whether the polar contacts stem from backbone (yellow) or side chain (purple) residues *on the peptide*. Peptide residues with frequent side chain interactions are labeled in red. **(C, E)** Residues outside of the binding β-strand that are important interaction sites shown in (C) are labeled in (E).

proximal to one another (namely ASCIZ/ATMIN, and BSN), which would facilitate binding of weaker motifs because of bivalency. Within the nonbinders, three of the four well-scoring nonbinders are listed in Fig 4D as "other," indicating that there is consistency between algorithm predictions and our ability to recognize binders/nonbinders based on sequence. This also suggests that there are some deleterious interactions that we have yet to understand and will require more data to decipher. The fourth sequence contains a hydrophobic valine at the +2 position (Fig 6B, sequence 8), which is very rare, as this position is often fully solvent exposed and prefers β-strand breaking residues (Fig 1B). Although LC8Pred weights valine at +2 negatively (Fig 6A), the remaining residues score well enough to result in the erroneous categorization of this sequence as a binder. Further accumulation of LC8-binding and nonbinding sequences will no doubt help to clarify the

importance of one poorly scoring residue and improve LC8Pred accuracy. Our LC8 motif algorithm is available on the database web page for public use (http://lc8hub.cgrb.oregonstate.edu/LC8Pred.php) for any sequence of interest.

### Predictive scores for the human protein Chica: a known LC8 binder

To test the ability of LC8Pred to identify binding sequences, we scored a test protein on each matrix using a sliding window. For this test, we selected Chica, a protein that contains a series of LC8-binding sequences between residues 400 and 475 (Clark et al, 2016). To prevent algorithmic bias, peptides from Chica were not used in the development of our scoring matrix. Upon applying the LC8Pred algorithm, six positive scores were returned within Chica (Fig 6C).

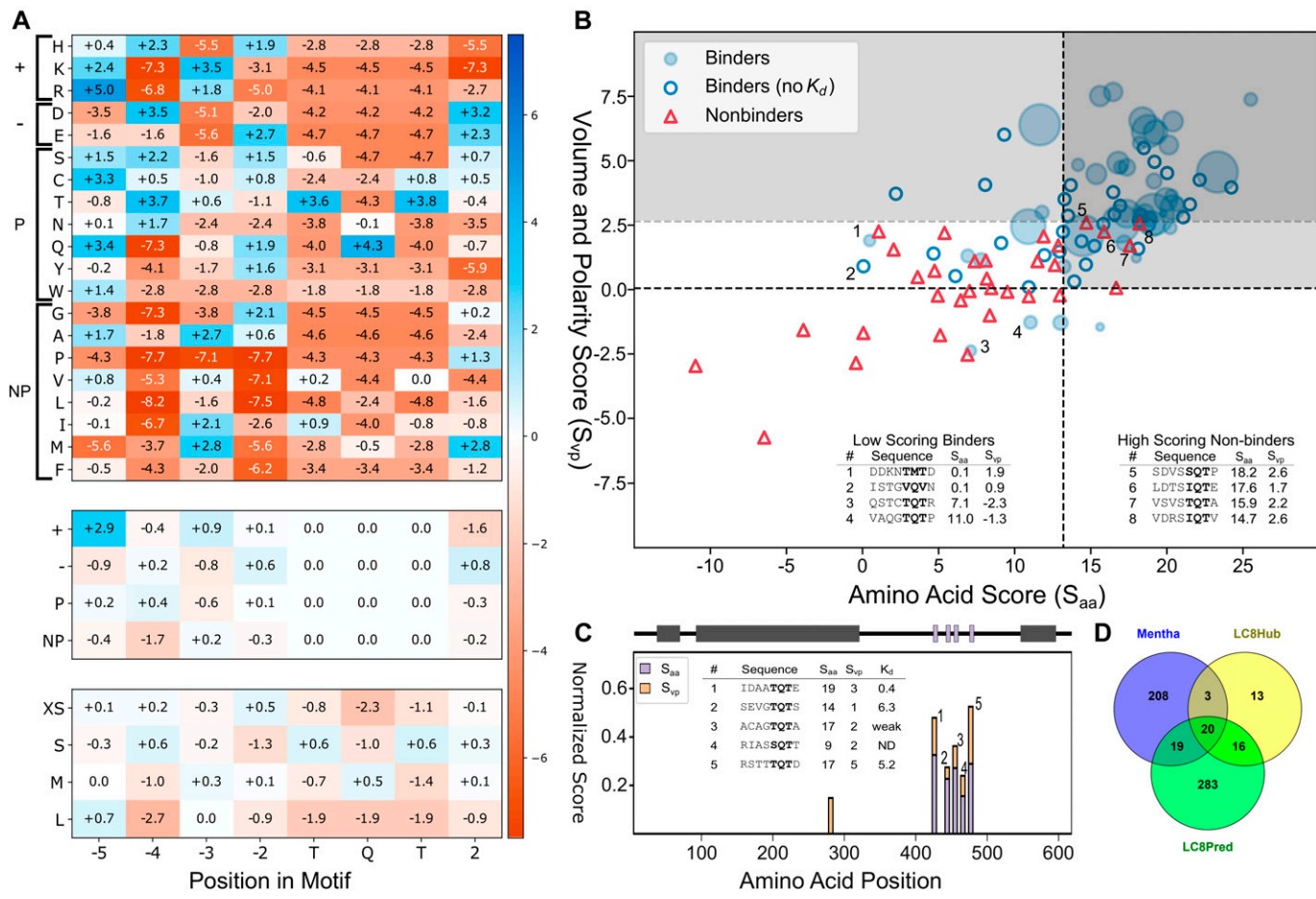

**Figure 6. Generation and testing of The LC8Pred algorithm.**
**(A)** PSSMs for amino acids (A, top), bins by chemical property—positively charged, negatively charged, polar, or nonpolar (middle), and bins by volume—less than 106 Å³, 122 to 142 Å³, 155 to 171 Å³, and greater than 200 Å³ (bottom). Values correspond to the combined weight at a given position for the binder-only matrix and the nonbinder-normalized matrix. **(B)** Scatterplot of available sequences scored using a leave-one-out method of cross validation. For binders with a known $K_d$, the size of the bubble was varied inversely with the $K_d$, with binders with a $K_d$ below 0.5 μM represented as the maximum possible dot size. Binder sequences with an unknown binding affinity were plotted as hollow circles and nonbinders as red triangles. The light grey box denotes predicted binding sequences using this scoring system. A second threshold for the volume and polarity axis indicates the very high confidence region, above which the specificity is unity. Outliers are noted in the tables (inset) and numbered in figure. **(C)** Normalized scores from matrices used to evaluate known LC8-binding protein Chica, where a score of one equates to the ideal amino acids of physicochemical properties at all positions. A sliding window to evaluate Chica for predicted binding sites across the protein was used, with the "0" position within the motif plotted (i.e., at 400, the 0 position is the 400th amino acid within Chica). A diagram of Chica showing secondary structure prediction (grey) and LC8 binding sites (purple) is above, and sequences predicted to bind are on the right, along with their corresponding scores. **(D)** Venn diagram of human proteins in the LC8Hub database, proteins that contain at least one LC8-binding sequence as determined by LC8Pred, and proteins reported to bind LC8 in the protein–protein interaction database Mentha (Calderone et al, 2013).

One of these scores fell far below threshold and was ignored. The remaining five scores were within the LC8-binding region; four of which have previously been determined experimentally to bind LC8 (Clark et al, 2016). The other is an SQT-containing sequence that scored below the designated threshold in the amino acid matrix, indicating that although this particular sequence may bind LC8, the prediction is of low confidence (Fig 6C). These test results provide strong evidence of the discriminatory power of our algorithm, as it can successfully recognize sequences that bind LC8 while excluding those that do not.

### Human proteome scan identifies 374 potential binding sequences

After determining LC8Pred's reliability and ability to distinguish potential motifs, we used it to scan the human proteome to identify high-confidence binding partners. In total, 785 sequences scored above our PSSM thresholds. These sequences were then further filtered using IUpred to eliminate motifs within ordered regions. This process yielded 374 high-confidence hits from 338 proteins (Table S2). Of these, 36 have been previously described in direct interaction studies and are listed on our LC8Hub database (Fig 6D). A further 19 partners have been identified in high throughput proteomics studies, such as pull-down mass spectrometry, including the highest scoring hit (FAM117B; Hein et al, 2015; Boldt et al, 2016). Our data validate these interactions and define likely binding regions within these partners. It is of note that several of the identified partners contain multiple putative LC8 sites in close succession. The ability of LC8 to "zip up" partners with multiple recognition motifs has been described for both Nup159 (Nyarko et al, 2013) and ASCIZ (Clark et al, 2018), and it is possible that many partners within this list contain weaker LC8 sites proximal to these tight-binding motifs.

Prior studies on LC8 interactions have noted an enrichment in LC8 partners within the Hippo signalling pathway (Erdős et al, 2017). Our proteome scan has identified these same partners (e.g., AMOT, WWC1, and WWC2) and additional novel binders from the hippo pathway, such as STK4 and DLG5. Interestingly, this pathway is the only "biological process" significantly enriched in LC8 binding partners, based on gene ontology analysis using the WebGestalt program (Wang et al, 2017).

To verify that LC8Pred is correctly predicting partners, we synthesized three peptides from Table S2 and tested their capacity to interact with LC8 via ITC. The three peptides were derived from the human proteins: HIV Tat-specific factor 1 (HTATSF1), a cofactor required for the Tat protein activation of human immunodeficiency virus transcription; otoferlin (OTOF), a calcium ion sensor involved in vesicle-plasma membrane fusion and neurotransmitter release, associated with hearing loss; and ninein (NIN), a component of the core centrosome and a dynein activator protein. These peptides were selected based on their mid-level scores and lack of prior data detailing LC8 interactions (Table S2). All three peptides bound to LC8, although only HTATSF1 was a "strong" binder with a fittable thermogram ($K_d$ of 10 $\mu$M). These data support the effectiveness of our LC8Pred algorithm and demonstrate that it is capable of predicting binding partners of varying affinities despite noncanonical motifs (Table S3).

# Discussion

Hub proteins are essential for cell viability as they are central in protein–protein interaction networks. Dynamic hubs such as LC8 often have a recognizable binding motif, which should allow for the prediction of binding partners without the need for exhaustive testing of each individual interaction (Madeira et al, 2015); however, no such program is available for LC8. Instead, binding partners are often identified via high-throughput pull-down experiments. For example, the interaction between LC8 and OFD1 was initially identified via pull-down mass spectrometry study in cilia (Boldt et al, 2016). In most cases, follow-up experiments for validation of direct binding are not performed, as it is prohibitively expensive to verify these interactions in a systematic fashion. Here, we validate purported and previously unreported LC8 binding partners (including OFD1), measure their binding affinities and thermodynamic properties, and establish a database of known LC8–partner interactions to define and describe generalizable requirements for LC8 motif recognition. We use these rules, along with amino acid preferences in nonbinding sequences, to develop an algorithm that effectively distinguishes between binding and nonbinding sequences, with the aim of facilitating a priori prediction and discovery of LC8–partner interactions with much greater confidence and accuracy than has been possible before now. Furthermore, we validate interactions that reinforce the importance of LC8 within a wide variety of systems and demonstrate that LC8 is both localized ubiquitously throughout the cell and enriched in distinct regions unrelated to the dynein complex.

Of the 72 synthesized tetradecameric peptides, we verified binding for 29 peptides derived from 27 distinct proteins (Table S3). Of these 27 proteins, 19 are newly identified LC8 binding partners. It is of note that all of our validated sequences contain the canonical TQT anchor (or variation thereof) at the C-terminus of the peptide, supporting the idea that a C-terminal anchor is vital for LC8 binding. Although the LC8 binding site is structurally dynamic, there are distinct preferences and exclusions for each position within the binding motif (Fig 4). In addition to the presence of an anchor, binders often have −4 positions capable of H-bonding, larger positive side chains at −3 positions, and strand breaking +2 positions. However, the presence of pre-anchor prolines, a high concentration of charges, or bulky hydrophobic groups at the −2 position will each limit the likelihood that a sequence will bind LC8 (Fig 4).

Algorithms for motif identifications have been developed for both 14-3-3 and calmodulin to efficiently predict potential binding partners. In the case of calmodulin, its diverse set of binding motifs has led to multiple programs (Yap et al, 2000; Mruk et al, 2014; Abbasi et al, 2017), which predict potential binding partners via a mixture of sequence similarity to known binders, $\alpha$-helical propensity, or the number of canonical calmodulin-binding motifs within a given sequence. In the case of 14-3-3, which binds phosphorylated sequences within disordered segments of proteins, the algorithm makes use of support vector machines and artificial neural networks (Madeira et al, 2015) and scores potential binding sequences using a PSSM. Here we succeeded in generating LC8Pred, an algorithm with a 78% accuracy rate, by incorporating nonbinder data and by reducing the PSSM dimensionality from 20 amino acids to four physicochemical categories, based on either polarity or volume. We have tested LC8Pred on the known LC8 binder Chica and by scanning the human proteome. In case of Chica, LC8Pred efficiently recognized known binding sites and excluded all other regions (Fig 6C). Our proteome scan identified 338 potential LC8 binding partners, including 19 binding partners that have been identified previously via high-throughput proteomics studies (Fig 6D and Table S2), providing a new set of high-confidence LC8-interacting proteins. Three peptides were selected from these potential partners and shown to indeed bind LC8.

The ability to bind a wide variety of sequences despite an extremely conserved binding interface is a hallmark of dynamic hubs, as exemplified by calmodulin (Frederick et al, 2007) and 14-3-3 proteins (Johnson et al, 2010). Crystal and NMR structures for LC8 show that the $\beta$3 strand at the partner binding interface has the highest sequence conservation (Fig 5A), and surprisingly, it is also the most dynamic region (Fig 5B). Consistent with the dynamic nature of the binding grooves, thermodynamic analyses of tight binding sequences demonstrate a wide range of entropy/enthalpy compensation, including some sequences that bind with a favorable change in entropy, such as ICE1 and VP4. Previous studies on LC8 dynamics of binding to dynein IC and the protein swallow (Swa) show that increases in ordered structure upon binding are peptide dependent (Hall et al, 2008). With Swa, the complex is more compact, rigid, and homogeneous than with IC, indicating that the IC peptide retains more freedom of motion in the bound state than does the Swa peptide. Consistent with these observations, IC binds with a favorable entropy, whereas Swa does not. Our work here demonstrates that these different modes of binding are not limited to IC and Swa but rather that entropic factors commonly modulate LC8 binding to accommodate extraordinary variation in binding sequences.

Hub proteins like LC8 are essential for cell homeostasis as they sit at the center of complex interaction networks; therefore, it is imperative to understand the rules that govern hub protein interactions. The dynamic nature of the LC8 pocket, and entropic contributions to binding, make it difficult to predict partners with high confidence, and yet it is this very dynamic characteristic that makes LC8 such a powerfully effective hub protein. Here we have amalgamated our experimentally verified LC8-binding sequences with all previously described binding sequences and developed an algorithm that significantly advances our ability to predict LC8 partners based solely on sequence. Confidence in a potential LC8-binding sequence can be further improved by considering the structure and conservation of the binding site, and we have therefore linked LC8Pred to ProViz, a tool that analyzes protein structure and conservation. In addition, it is important to note that LC8Pred is optimized for stringency and predicting tight binding interactions and does not account for adjacent oligomerization sites, which would increase binding affinities. Future versions of the algorithm will incorporate parameters to account for other factors impacting binding, such as oligomerization state or subcellular localizations. We also anticipate that the predictive power of our algorithm will improve dramatically as more LC8-binding and nonbinding sequences are identified and deposited in the LC8hub database, resulting in a comprehensive view of the LC8 hub interaction network.

## Materials and Methods

### Cell culture and transfection

HeLa Kyoto cells stably expressing LC8-GFP were a kind gift from I Poser and A Hyman (Max Planck Institute of Molecular Cell Biology and Genetics, Dresden, Germany) (Poser et al, 2008). The cells were cultured in DMEM with 10% (vol/vol) fetal calf serum and with 1% (vol/vol) penicillin/streptomycin. The cell line was routinely checked for mycoplasma contamination using LT07-518 Mycoalert assay (Lonza). The identity of the cell line was monitored by immunofluorescence staining–based analysis with multiple markers.

### Microscopy and image analysis

Live and fixed samples were imaged with spinning disk microscopy, which was performed on an inverted research microscope Eclipse Ti-E with the Perfect Focus System (Nikon), equipped with Nikon Plan Apo VC 100× N.A. 1.40 oil objective, Yokogawa CSU-X1-A1 spinning disc, Vortran Stradus 405 nm (100 mW), Cobolt Calypso 491 nm (100 mW) and Cobolt Jive 561 nm (100 mW) lasers, Chroma emission filters ET460/50m (part of 49021 filter set), ET525/50m (part of 49002 filter set) and ET630/75m (part of 49008 filter set), ASI motorized stage MS-2000-XYZ with Piezo Top Plate (ASI), Photometrics Evolve 512 EMCCD camera (Photometrics), and controlled by MetaMorph 7.7 software (Molecular Devices). Images were projected onto the camera chip with intermediate lens 2.0× (Edmund Optics) at a magnification of 0.067 mm/pixel. To keep cells at 37°C, we used the stage top incubator INUBG2E-ZILCS (Tokai Hit). Z-series of live and fixed samples were acquired using a 0.1-$\mu$m-step confocal-based scan. Side views were reconstructed by projecting maximum fluorescence intensities of 24 X 12-$\mu$m side view slices.

Alternatively, fixed samples were imaged using wide-field fluorescence illumination on a Nikon Ni upright microscope equipped with DS-Qi2 camera (Nikon), Intensilight C-HGFI illuminator (Nikon), ET-DAPI, ET-EGFP and ET-mCherry filters (Chroma), Nikon NIS Br software, and a Plan Apo Lambda 100× oil NA 1.45 (Nikon) objective. For presentation, images were adjusted for brightness and contrast using ImageJ 1.47v (NIH).

### Localization prediction

Localization information is derived from the COMPARTMENTS program (Binder et al, 2014), using the curated "Knowledge-based" evidence category. The list of LC8 binding proteins used matches the curated list on the LC8 database, described in this article. Only data with confidence scores of three or higher (out of five) are included. Cellular compartments are simplified for depiction purposes (e.g., "other organelles" includes Golgi bodies, mitochondria, and so on).

### ProP-PD selections

Phage display selections were performed using a proteomic library designed from the disordered regions of the human proteome described in the study by Davey et al (2017). Selections were performed with minor adjustments. GST-LC8 (0.1 mg/ml in 100 $\mu$l TBS, 50 mM Tris–HCl, 150 mM NaCl, pH 7.4) was coated on a Maxisorp 96-well plate (Nunc) via overnight shake-incubation at 4°C. Plates were blocked with 0.5% BSA in TBS for 1 h at 4°C and washed with TBS. The phage library was added to the well (100 $\mu$l) and incubated for 2 h at 4°C. Unbound phages were removed by washing plates five times with 300 $\mu$l TBS + 0.05% Tween. Bound phages were eluted by infection into 100 $\mu$l log-phase *Escherichia coli* Omnimax cells (Invitrogen; OD: 0.3–0.8) in 2xYT media (10 g bacto-yeast extract, 16 g bacto-tryptone, 5 g NaCl per liter) supplemented with 10 $\mu$g/ml tetracyclin. After a 30-min shake-incubation at 37°C, the bacteria were hyperinfected with M13K07 helper phages for 45 min to allow phage production. Cultures were transferred into 5 ml 2xYT, 0.3 mM IPTG, and grown overnight with antibiotics (25 $\mu$g/ml kanamycin and 100 $\mu$g/ml carbenicillin). The bacteria were pelleted by centrifugation. 1 mL of the phage supernatant was extracted and heat inactivated at 65°C for 20 min. Finally, the solution was pH neutralized using 10× TBS, and the phage pool was used in the next round of selection. Five rounds of phage selections were performed in total. The phage pool from the fourth day of selection was used for clonal phage ELISAs and sequencing. For next-generation sequencing, 5 $\mu$l of the phage pool from the fourth day of selection was used as template in a barcoding PCR. The sample was prepared and analyzed as described in detail elsewhere (Wu et al, 2017).

### Peptide synthesis

A total of 72 putative binding partners identified from ProP-PD selections and algorithm predictions were commercially synthesized from either Genscript, or Synpeptide, as 14–16 amino acid

sequences. Non-native residues were added to the termini of some peptides to facilitate solubility and peptide concentration determination (Tables 1 and 2, italics). All peptides were derived from either human or viral proteins.

## ITC

ITC experiments for the interactions of LC8 with peptides were performed using a Microcal VP-ITC microcalorimeter at 25°C in buffer composed of 50 mM sodium phosphate, 50 mM NaCl, 1 mM sodium azide, and 5 mM $\beta$-mercaptoethanol, pH 7.5. Some peptides contained cysteine residues, so 5 mM $\beta$-mercaptoethanol was included in all solutions for consistency. In all experiments, an initial 2 $\mu$l injection was followed by 26–50 injections of 3–10 $\mu$l peptide (500 $\mu$M) into 25 $\mu$M LC8 in the sample cell. Number and volume of injections were adjusted for each experiment to minimize ambiguity in the shape behaviour of isotherms and thermograms. Peptide concentrations were determined from absorbances at 280 nm using molar extinction coefficient values computed with the Protparam tool on the ExPASy website (Gasteiger et al, 2005). Peptides lacking aromatic residues were weighed and resuspended in the proper volumes to ensure 500 $\mu$M final concentrations. Protein samples and buffer were degassed before data collection. Data were processed using Origin 7.0 (Microcal) and fit to a single-site binding model. Final values for binding parameters are averages of two to three independent experiments.

## LC8Pred algorithm generation

The LC8Pred algorithm was developed using 79 LC8 binding sequences and 32 anchor-containing nonbinding sequences (Table S4). We selected sequences that bind LC8 with high confidence, on which direct interaction data are available. In addition, all sequences with a $K_d$ above 25 $\mu$M were not included. The TQT (or variation thereof) anchor-containing nonbinders were those peptides shown by ITC to have no binding to LC8.

In addition, a new series of matrices were developed which binned amino acids into categories based on physicochemical properties. Specifically, a matrix that separates amino acids into positively charged, negatively charged, hydrophobic, or polar and uncharged, and a matrix that separates amino acids into four groups based on volume, with volume bins being selected to minimize the range of volumes within each bin. We built these matrices to overcome the limitation of our small dataset, as reducing the number of groups from 20 amino acids to four possible properties improves the likelihood that some information is available for a given position and a given property within the motif.

In total, six matrices were developed, two for each set of bins (amino acid, polarity, and volume). For a given bin, one matrix was normalized to the background frequency of a given amino acid or a given property within the disordered eukaryotic proteome taken from the DisProt database of intrinsically disordered regions (Piovesan et al, 2017). For the other matrix, normalization was done for the frequency of a given amino acid or property in the nonbinder dataset. As nonbinding sequences were selected based on the presence of an anchor, there is no enrichment or depletion at the

anchor positions of −1 to +1. These positions were therefore ignored in these matrices.

To simplify our scoring system, we combined the matrices into two simple scoring metrics, $S_{aa}$ and $S_{vp}$, where $S_{aa}$ is a combination of the two matrices that use amino acid–type bins, and $S_{vp}$ is a combination of the four matrices that use volume or polarity bins. To determine how effective each individual matrix was at separating binding and nonbinding sequences, we scored our available sequences using leave-one-out cross validation, where a given sequence was excluded from the matrix and then scored. The leave-one-out approach was used to combat the difficulty of our limited dataset.

We used receiver operating characteristic (ROC) curves (Supplemental Data 1) as a metric of the effectiveness of each score. The area under these curves corresponds to the ability of each matrix to separate binding sequences from nonbinding sequences. We then combined scores into the $S_{aa}$ and $S_{vp}$ scores described above, where each individual matrix score was weighted through a grid search of possible weights, where the largest area under the ROC was taken to be the optimal weight for each score. Surprisingly, the area under the ROC curve was highest when the binder-only polarity matrix was removed from the $S_{vp}$ score. Positions −1, 0, and 1 are therefore not weighted in the polarity matrix (Fig 6A) because the nonbinder normalized matrix was also excluded at those positions because of a lack of anchor enrichment, as discussed above.

## The LC8 motif repository

We have manually curated a database that compiles information for all known LC8 binding partners. Including the 19 binding partners identified in this work, there are currently 80 experimentally confirmed LC8 interacting partners containing 116 individual anchor motifs. Of these binding motifs, 98 have been confirmed by in vivo or in vitro experiments, with a further 18 identified through biochemical screening methods. The database serves to (1) provide a source of up-to-date information on LC8 and its cellular role, (2) organize and classify LC8 binding proteins in an easily searchable manner, and (3) list the sequences of all TQT motifs to aid in identification of new binding partners. Access to the motif repository is available at http://LC8hub.cgrb.oregonstate.edu. For each protein, the following information is provided: the species, TQT peptide sequence, number of motifs in the protein, Protein Data Bank (PDB) ID (if a structure exists), reference link, and interaction type. The interaction type has three levels of classification, depending on the method by which the LC8–partner interaction was identified: (1) high-throughput biochemical method, such as yeast-2-hybrid, where the interaction has not been confirmed by in vivo or in vitro experiments; (2) in vivo experiments, such as mutation or knockout experiments, where a function for the LC8-partner complex has been identified; and (3) in vitro experiments that determine the binding affinity, structure, or other information about the LC8–partner interaction. In addition, sequences of interest can be tested at LC8Hub by inputting a .fasta file or a string of letters corresponding to the protein sequence of interest. Output provides both the $S^{aa}$ and the $S^{vp}$ scores, and indicating sequences that are likely to bind LC8 according to available data. Finally, sequences determined to

either bind or not bind LC8 despite the presence of an anchor sequence can be submitted for incorporation into the database. It is our hope that the information in this database will facilitate research on LC8 and, by enhancing our understanding of the TQT motif, enable more robust prediction of new binding partners.

### Structure and motif analysis

Structures of LC8 were obtained from the PDB (free LC8 PDB codes: 1PWJ, 1PWK, 1RE6, 3BRI, 5WOF; bound to peptides: 2XQQ, 4QH7, 3E2B, 2P2T, 3BRL, 3DVP, 3P8M, 3ZKE, 4D07, 4HT6, 5E0M). All images were generated using PyMol (Schrodinger, 2010). Peptides without structures available were built in silico using Chimera (Pettersen et al, 2004). Peptides in Fig 4 are colored according to enrichment and depletion tables for amino acids, shown in Fig 6A (blue for scores >1, white for scores between 1 and −1, and red for scores <1). Solvent accessible surface area analysis was performed using a representative LC8 crystal structure (2XQQ) with the GETAREA program (Fraczkiewicz & Braun, 1998). Protein charge potential was calculated for LC8 using PyMol's built-in charge-smoothed potential calculator. Two-dimensional lig-plots were generated using ChemDraw.

Alignment of LC8 structures was done using the Ensemblator (Brereton & Karplus, 2018) program, and the RMSD for residues in free LC8 structures (listed above) was calculated using the built-in local alignment tool. This tool works by aligning each dipeptide within the protein and calculating the RMSD for the next amino acid within the protein sequence. A representative structure was then colored based on these values to demonstrate structural conservation. Sequence-based conservation was performed using ConSurf (Ashkenazy et al, 2016), with LC8 sequences from 58 different eukaryotic species.

## Supplementary Information

## Acknowledgements

This work was supported by the National Science Foundation grant MCB 1617019 to E Barbar, by the Swedish Research Council (C0509201) to Y Ivarsson, by Lennanders foundation and Ingegerd Bergh's foundation to C Blikstad. The NGS was performed in collaboration with Eduard Resch, Fraunhofer Institute for Molecular Biology and Applied Ecology IME TMP, Frankfurt am Main, Germany. Y-C Ammon is supported by the MARIE SKŁODOWSKA-CURIE ACTIONS Innovative Training Network 675407 PolarNet.

### Author Contributions

N Jespersen: data curation, formal analysis, investigation, and writing—original draft, review, and editing.
A Estelle: data curation, software, formal analysis, methodology, and writing—original draft.
N Waugh: data curation.
NE Davey: formal analysis and methodology.
C Blikstad: data curation, investigation, and writing—review and editing.
Y-C Ammon: data curation.
A Akhmanova: data curation, supervision, and writing—review and editing.
Y Ivarsson: data curation, formal analysis, supervision, funding acquisition, and writing—original draft, review, and editing.
DA Hendrix: supervision, methodology, and writing—review and editing.
E Barbar: conceptualization, resources, supervision, funding acquisition, project administration, and writing—original draft, review, and editing.

### Conflict of Interest Statement

The authors declare that they have no conflict of interest.

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
