## [Reviewer comments · Life Science Alliance]

Life Science Alliance

Systematic identification of recognition motifs for the hub protein LC8

Nathan Jespersen, Aidan Estelle, Nathan Waugh, Norman Davey, Cecilia Bilkstad, York-Christoph Ammon, Anna Akhmanova, Ylva Ivarsson, David Hendrix, and Elisar Barbar

DOI: <https://doi.org/10.26508/lsa.201900366>

Corresponding author(s): Elisar Barbar, Oregon State University

Review Timeline:

Submission Date:	2019-03-05
Editorial Decision:	2019-04-08
Revision Received:	2019-05-24
Editorial Decision:	2019-06-19
Revision Received:	2019-06-21
Accepted:	2019-06-24

Scientific Editor: Andrea Leibfried

Transaction Report:

April 8, 2019

Re: Life Science Alliance manuscript #LSA-2019-00366-T

Prof Elisar Barbar
Oregon State University
2011 ALS
Corvallis, OR 97331

Dear Dr. Barbar,

Thank you for submitting your manuscript entitled "Systematic identification of recognition motifs for the hub protein LC8" to Life Science Alliance. The manuscript was assessed by expert reviewers, whose comments are appended to this letter.

As you will see, the reviewers appreciate your work but think that you may have missed a lot of true interactors in the validation assays in case of weak binding. Reviewer #1 further notes that the value provided would be much stronger if some or at least one novel, algorithm-based hit gets tested further. We would like to invite you to provide a revised version of your work, addressing these issues. Both reviewers provide constructive input on how to validate weak interactions and how to extend the resource value provided by further proof-of-principle analyses.

Thank you for this interesting contribution to Life Science Alliance. We are looking forward to receiving your revised manuscript.

Sincerely,

B. MANUSCRIPT ORGANIZATION AND FORMATTING:

Reviewer #1 (Comments to the Authors (Required)):

LC8 is a small dimer that binds a large number of proteins in many different pathways. A key step to understanding where it plays a role will be identifying which proteins it can interact with. This is a challenging problem as there is a large flexibility in the sequences it binds.

The authors provide a phage display approach to identify new interacting proteins. They identify binders and non-binders among the candidates using an ITC approach. This leads them to identify a series of rules for LC8 binding proteins that they develop into a web-based algorithm. They run their program on the human proteome and identify candidate interacting proteins. Some of these are already known LC8 binders.

The manuscript addresses an important problem. It generates a useful tool for those studying pathways involving LC8. My concerns are as follows:

- 1) The authors appear to define non-binders as proteins that were identified by phage display, but showed no interaction in ITC experiments with isolated peptides. Is it not probable that many of these "non-binders" are in fact weak binders? This would mean that the rules the authors develop are likely to miss many potential interaction sites?
- 2) The authors discuss the fact that LC8 interactions are often increased by a bivalent effect. I wonder if they had considered performing ITC with peptides that are linked together (e.g. by fusion to GST). Such an approach might mimic the bivalent effect and allow them to identify the weaker interactions.
- 3) The paper would be made much stronger by testing at least some of the human proteome hits identified by the algorithm. I realize this is a lot of work and may be outside the scope of the paper.

Reviewer #2 (Comments to the Authors (Required)):

This manuscript outlines work on the short linear motif (canonically RxTQT) which binds to the LC8 protein. LC8 was originally described as a dynein light chain subunit. It has turned out to have more diverse roles and is involved in oligomerising many proteins and protein complexes via this motif. Thus LC8 is an important protein to address for the understanding of cell regulation. However, it has not proven entirely straightforward to describe the sequence variability allowed for the motif. In the present work, a phage display library of the human "disorderome" was probed against LC8. Peptides corresponding to binders were then synthesised and tested with ITC for binding and affinity measurements. Based on positive and negative binders, a method for predicting LC8-binding peptide candidates has been developed. In addition, a database of LC8 interactors has been collated from the literature. Overall, this work is a useful contribution to LC8 research. There are a number of issues that the authors might consider for improving the manuscript.

The LC8 database and prediction tool is not fully available on-line. It should have been available for review and definitely must be available before a revised manuscript can be considered.

A majority of the peptides retrieved by phage display did not turn out to bind when tested in solution. The paragraph addressing this point on P. 5 is insufficient. In particular, many of the sequences in SupTable 1 have a TQT core that is close to the peptide N-terminus. This means that the actual binding sequence includes several phage protein residues. This point and perhaps other reasons that the authors might suspect to be involved in the unusually high failure rate should be discussed.

When presenting proteins like Chica give the species that the protein came from.

Emphasise in the discussion the importance of sequence conservation for motif detection as a step that users of the prediction tool should always undertake.

Fig. 3A. Name the peptides used for the ITC plots shown.

Table 2. Line up the peptide sequences by their TQT cores.

In SupTable 1, MUC4 has an alternative motif candidate (RESQT) to the one shown.

In SupTable 3, Use courier for the sequence column

ELISAR BARBAR, PhD
DEPARTMENT OF BIOCHEMISTRY AND BIOPHYSICS
OREGON STATE UNIVERSITY
2011 Agriculture and Life Sciences · Corvallis, Oregon 97331-7305 USA
Telephone 541-737-4143 Fax 541-737-0481

May 24, 2019

Dear Dr. Leibfried,

Thank you for your letter of Apr 8th, 2019, regarding the manuscript LSA-2019-00366-T and your favorable review. Below we have addressed point by point all the issues raised by both reviewers. Briefly, we agree with the reviewer that we “may have missed a lot of true interactors in case of weak binding”, but as we explain below, the goal is not to identify new partners but to identify tight binding partners that can be used to develop a robust algorithm that predict binders. Regarding the second point “the value provided would be much stronger if some or at least one novel, algorithm-based hit gets tested further”. We have acted on the reviewer’s recommendation and synthesized three peptides corresponding to novel algorithm-identified partners, and shown that they indeed bind LC8, confirming the predictive power of the algorithm.

Reviewer #1 (Comments to the Authors (Required)):

1) The authors appear to define non-binders as proteins that were identified by phage display, but showed no interaction in ITC experiments with isolated peptides. Is it not probable that many of these "non-binders" are in fact weak binders? This would mean that the rules the authors develop are likely to miss many potential interaction sites?

We thank the reviewer for their insightful comment, and while we agree that it is indeed possible that some of our non-binding peptides may bind in other contexts, the affinities based purely on their sequence are significantly less than 25 μ M (the cut-off for well-fit data in our methodology). Sequences that showed evidence of binding, but were not fit were classified as weak binders (See Figure 3A, Table 1). The goal of this work is to identify what makes a sequence likely to bind LC8, and we have therefore built our algorithm using only tight binding sequences, and excluded those that are weak binders (See Methods) in the hopes that we would build a higher confidence predictor, rather than the more common tools which will always predict a binding site for a protein of interest. That said, the reviewer makes a good point about the possibility of us missing many weaker binding interactions with this experimental throughput, and we have therefore included a statement in the Discussion and Methods (Lines 493-497, Supplemental Information Lines 5-7) noting that our algorithm will surely miss many weak-interacting partners, particularly those that bind adjacent to oligomerization domains or within multivalent LC8-binding regions. Indeed, it could be said that our algorithm is designed to look at specifically “dimer-inducing” sequences, which may only represent a subset of all LC8 binding motifs.

2) The authors discuss the fact that LC8 interactions are often increased by a bivalent effect. I wonder if they had considered performing ITC with peptides that are linked together (e.g. by fusion to GST). Such an approach might mimic the bivalent effect and allow them to identify the weaker interactions.

This is a very interesting idea, and gets at a question that is an active area of study in the Barbar lab, *i.e.* the role of bivalency/multivalency in LC8 interactions; however, as stated above, the goal of this work was to predict binding partners based purely on sequence, and have therefore *not* included any information that can be gleaned from regions outside of the binding motif. In future iterations of the algorithm we hope to include parameters which would account for a bivalency ‘boost,’ but we believe that this is outside of the scope of this paper.

3) The paper would be made much stronger by testing at least some of the human proteome hits identified by the algorithm. I realize this is a lot of work and may be outside the scope of the paper.

At the reviewer's suggestion, we have ordered and tested binding of 3 peptides of interest, and demonstrated that all 3 bind (Supplemental Table 2,3; Lines 414-423). These peptides were tested via ITC, as described for previous peptides, and demonstrate the functionality of our algorithm.

Reviewer #2 (Comments to the Authors (Required)):

The LC8 database and prediction tool is not fully available on-line. It should have been available for review and definitely must be available before a revised manuscript can be considered.

The database and prediction tool are now publicly available at <http://lc8hub.cgrb.oregonstate.edu/LC8Pred.php>. Inquiries can be submitted as a sequence of amino acids, a document with a set of FASTA formatted sequences, or a Uniprot code. Additionally, based on the reviewer's recommendation, we have linked LC8Pred to the protein alignment tool ProViz, which will allow the user to look at the disorder and conservation of sequences of interest.

A majority of the peptides retrieved by phage display did not turn out to bind when tested in solution. The paragraph addressing this point on P. 5 is insufficient. In particular, many of the sequences in SupTable 1 have a TQT core that is close to the peptide N-terminus. This means that the actual binding sequence includes several phage protein residues. This point and perhaps other reasons that the authors might suspect to be involved in the unusually high failure rate should be discussed.

We thank the reviewer for their astute observation and recommendation. This point was previously discussed in the Supplemental Information section "Peptide Design;" however, as noted by the reviewer, this is a large factor in the success rate of our Phage display experiment, and we have therefore moved it to the main body (Lines 214-234).

When presenting proteins like Chica give the species that the protein came from.

With the exception of the viral proteins, all sequences discussed in this work are for human isoforms, as the Phage display used a library of disordered peptides in human proteins. We have therefore included a statement in the Methods indicating this (Lines 561-562), and have additionally edited the Chica section to note its human origin (Line 383).

Emphasize in the discussion the importance of sequence conservation for motif detection as a step that users of the prediction tool should always undertake.

This is a very important point, and we appreciate the reviewer's recognition of this fact. We have added a statement to the discussion propounding the importance of conservation predictions (Lines 491-493), and included a direct link to a pre-submitted alignment tool on the LC8Pred website in order to make it clear that best practice is to consider conservation when predicting LC8 binding sites.

Fig. 3A. Name the peptides used for the ITC plots shown.

Table 2. Line up the peptide sequences by their TQT cores.

In SupTable 3, Use courier for the sequence column

We thank the reviewer for their clarifying recommendations, and have edited the figures and/or legends accordingly.

In SupTable 1, MUC4 has an alternative motif candidate (RESQT) to the one shown.

We appreciate the careful review of our manuscript. Muc4 (and indeed, all mucins) are a very interesting case of multivalent LC8 binders, as they have multiple potential sites in close proximity. In fact, Mucin-16 comes up in Supplemental Table 2 more frequently than any other protein, indicating it may have as many as 10 LC8 binding motifs. The peptides we tested were based on the pull-down data, but then shifted according to what we believed to be the most likely sequences (See lines 214-234), but it is totally possible that the RESQT sequences described above could bind instead. That said, the MUC4

peptide we tested *did* bind, indicating that that sequence has the necessary motif features to interact with LC8, and justifying our choice in peptides. In the future, analysis of the extreme enrichment of LC8 motifs within Mucin proteins could be a very interesting area of study, but we believe that commentary on that is outside of the scope of this work.

I trust that this revised manuscript is now suitable for publication in Life Science Alliance . We look forward to hearing your editorial decision.

Sincerely yours,

Elisar Barbar
Professor of Biochemistry and Biophysics

June 19, 2019

RE: Life Science Alliance Manuscript #LSA-2019-00366-TR

Prof. Elisar Barbar
Oregon State University
2011 ALS
Corvallis, OR 97331

Dear Dr. Barbar,

Thank you for submitting your revised manuscript entitled "Systematic identification of recognition motifs for the hub protein LC8". As you will see, the reviewers appreciate the introduced changes and we would be happy to publish your paper in Life Science Alliance pending final revisions necessary to meet our formatting guidelines:

- It is unclear how many cells were analysed in figure 2 (A and B, but also to arrive at the display in figure 2C) - please clarify and add this information to the figure legend
- please add panel labels to Figure 6
- Please incorporate the suppl information into the main manuscript file, the Tables S1 and S3 can get added to this docx file as well
- Please note that table S2 is called 'figure' instead of 'table' in the xlsx file
- Please list 10 authors et al in your reference list

A. FINAL FILES:

- An editable version of the final text (.DOC or .DOCX) is needed for copyediting (no PDFs).
- High-resolution figure, supplementary figure and video files uploaded as individual files: See our detailed guidelines for preparing your production-ready images, <http://www.life-science-alliance.org/authors>
- Summary blurb (enter in submission system): A short text summarizing in a single sentence the

study (max. 200 characters including spaces). This text is used in conjunction with the titles of papers, hence should be informative and complementary to the title. It should describe the context and significance of the findings for a general readership; it should be written in the present tense and refer to the work in the third person. Author names should not be mentioned.

B. MANUSCRIPT ORGANIZATION AND FORMATTING:

Sincerely,

Reviewer #1 (Comments to the Authors (Required)):

The authors have addressed my concerns. The addition of experimental data to test their predictions strengthens the manuscript and I recommend it for publication.

Reviewer #2 (Comments to the Authors (Required)):

The revised version of the LC8 motif paper has addressed my major concerns. I was now able to evaluate the LC8 database and LC8Pred servers and these worked well. Overall this paper is a useful advance for LC8 research and the prediction tools will be helpful for future motif discovery.

A minor point arising from the authors' reply point re the Mucins: The TQT motif matches are in the extracellular parts of these integral membrane proteins. If there is no known function for LC8 outside of the cell, then these might not be biologically meaningful and it should be borne in mind that large proteins inevitably have matches to numerous different SLiM patterns.

June 24, 2019

RE: Life Science Alliance Manuscript #LSA-2019-00366-TRR

Prof. Elisar Barbar
Oregon State University
2011 ALS
Corvallis, OR 97331

Dear Dr. Barbar,

Thank you for submitting your Research Article entitled "Systematic identification of recognition motifs for the hub protein LC8". It is a pleasure to let you know that your manuscript is now accepted for publication in Life Science Alliance. Congratulations on this interesting work.

DISTRIBUTION OF MATERIALS:

Again, congratulations on a very nice paper. I hope you found the review process to be constructive and are pleased with how the manuscript was handled editorially. We look forward to future exciting submissions from your lab.

Sincerely,
